# Accuracy of point-of-care SARS-CoV-2 detection using buccal swabs in pediatric emergency departments

Stephen B. Freedman,[1,2,3,4] Kelly Kim,[5,6] Gabrielle Freire,[7] Alicia Kanngiesser,[5,6] April Kam,[8,9] Quynh Doan,[10,11] Bruce Wright,[12,13] Maala Bhatt,[14] Simon Berthelot,[15] Jocelyn Gravel,[16] Brett Burstein,[17,18,19] Jason Emsley,[20] Ahmed Mater,[21,22] Robert Porter,[23] Naveen Poonai,[24,25,26] Deepti Reddy,[27] Richard J. Webster,[27] David M. Goldfarb,[28] Kirk Leifso,[29] Roger Zemek,[14,30] on behalf of Pediatric Emergency Research Canada (PERC) COVID Study Group

**ABSTRACT** To optimize the identification of severe acute respiratory syndrome coronavirus 2 (SARS-CoV-2)-infected children, specimen collection and testing method are crucial considerations. Ideally, specimen collection is easy and causes minimal discomfort, and the laboratory approach is simple, accurate, and rapid. In this prospective cohort study we evaluated the accuracy of a point-of care nucleic acid device using caregiver/patient self-collected buccal swabs. Participants were recruited in 14 Canadian tertiary care pediatric emergency departments. Children <18 years of age deemed to require SARS-CoV-2 testing were eligible. Caregivers or the patient-collected buccal swabs which were tested on the ABBOTT ID NOW. The reference standard was nasopharyngeal swab specimens collected by a healthcare provider tested via laboratory reverse transcription PCR (RT-PCR). We enrolled 2,640 study participants and 14.4% (381/2,640) were SARS-CoV-2 RT-PCR-positive. Eight percent (223/2,640) and 85.0% (2,244/2,640) were concordant test-positive and concordant test-negative, respectively. Sensitivity and specificity of the investigational approach were 58.5% [95% confidence interval (CI): 53.4, 63.5] and 99.3% (95% CI: 98.9, 99.6), respectively. Cycle threshold values were lower among concordant [median 17 (15, 21)] relative to discordant [median 30 (22, 35)] swabs ($P < 0.001$). Sensitivity was greatest among children <4 years of age, when caregivers performed the swabs, among unvaccinated children, and those with shorter symptom duration. Across multiple pain measures, less pain was associated with buccal swab testing. Although accuracy of the buccal swab point-of-care SARS-CoV-2 test was good and negative agreement was excellent, sensitivity was only 58.5%. Concordance was greater among those with higher viral loads, and the approach involving buccal swabs was less painful.

**IMPORTANCE** To optimize the identification of severe acute respiratory syndrome coronavirus 2 (SARS-CoV-2)-infected children, specimen collection and testing method are crucial considerations. Ideally, specimen collection is easy and causes minimal discomfort, and laboratory approach is simple, accurate, and rapid. We evaluated the accuracy and pain associated with buccal swab specimen collection by caregivers or children themselves who were tested using a point-of-care isothermal nucleic acid amplification SARS-CoV-2 test. This novel approach was compared to nasopharyngeal swab specimens tested using laboratory-based PCR tests. While negative agreement was excellent, positive percent agreement was less than 60%. Concordance was greater among those with higher viral loads, and thus, sensitivity is excellent when transmissibility is more likely to occur. Importantly, the approach involving buccal swabs was significantly less painful, and thus, children and their caregivers are more likely to agree to testing using such an approach.

**Peer Reviewer** James J. Dunn, Texas Children's Hospital, Houston, Texas, USA

Address correspondence to Stephen B. Freedman, Stephen.freedman@ahs.ca.

Dr Burstein reported receiving a career grant from Fonds de Recherche Quebec Sante during the conduct of the study. Dr Berthelot reported receiving a career grant from Fonds de Recherche Quebec Sante during the conduct of the study. Dr Freedman is supported by the Alberta Children's Hospital Foundation's Professorship in Child Health and Wellness. No other disclosures were reported.

See the funding table on p. 12.

**CLINICAL TRIALS**  Registered at ClinicalTrials.gov (NCT05040763).

**KEYWORDS**    COVID-19, point-of-care systems, diagnostic techniques and procedures, child, emergency service, hospital, pain

The collection of nasopharyngeal specimens remains the optimal approach to detect severe acute respiratory syndrome coronavirus 2 (SARS-CoV-2) in healthcare settings. However, nasopharyngeal specimen collection is uncomfortable [1], and this has contributed to a decreased willingness of the public, particularly children, to undergo testing [2]. In addition, nasopharyngeal swab testing must be performed by a healthcare provider [3], consuming resources and increasing costs [4].

To optimally detect SARS-CoV-2 in clinical laboratories, reverse transcription PCR (RT-PCR) testing is performed to detect viral RNA. This process is expensive and requires technical expertise and takes considerable time to process in the laboratory. Although we are no longer in the throes of the COVID-19 pandemic, testing continues to be performed for a variety of reasons including clinical diagnosis, screening, and public health surveillance. Moreover, we have also learned the importance of preparing for the next pandemic, and as such, there is a need to consider appropriate diagnostic approaches in advance of a future pandemic.

Saliva samples have been explored as a potential alternative to nasopharyngeal swabs [5], and such samples have a sensitivity, specificity, and average overall agreement of 87%, 99% [6], and 90%, respectively [7]. However, few studies included infants and toddlers, and among older children, specimens were collected using sponges [8]. Results of published buccal swab studies are inconsistent, and most reports do not differentiate "saliva," which cannot be obtained from very young children, from "buccal swabs [9–18].

Given the above, we conducted a study that compared the diagnostic accuracy of a SARS-CoV-2 point-of-care nucleic acid-based instrument using caregiver/patient self-collected buccal swabs with nasopharyngeal swabs collected by healthcare providers tested using RT-PCR in the clinical diagnostic laboratory. We also examined the acceptability and pain associated with buccal and nasopharyngeal swabs.

## MATERIALS AND METHODS

### Study setting and design

This prospective cohort study recruited participants between November 9, 2021, and October 15, 2022, in 14 Canadian tertiary-care pediatric emergency departments (ED) that are members of the Pediatric Emergency Research Canada (PERC) network [19]. Participating institutions obtained research ethics board approval; informed consent and participant assent (when appropriate) were obtained.

### Study participants

Eligible participants were <18 years of age, presented to a participating ED, and underwent nasopharyngeal SARS-CoV-2 testing at the discretion of the treating physician, based on the presence of symptoms or epidemiologic risk factors. All participants had their standard-of-care nasopharyngeal swab tested in accordance with local laboratory RT-PCR standards. Standard-of-care testing was performed in accredited clinical microbiology laboratories providing services to tertiary care pediatric institutions using either commercially available real-time RT-PCR assays or laboratory-developed assays with similar equivalent analytical sensitivities [20]. Potentially eligible participants were consecutively approached by a member of the research team; ED research coverage varied by site and day of the week. Children were excluded if they were unable or unwilling to provide a buccal swab specimen or were unable to communicate in English or French, preventing informed consent. Those with swab results reported as "indeterminate" via either testing method were excluded. Results of the standard-of-care nasopharyngeal swab were unavailable at the time of recruitment.

## Investigational device

The ABBOTT ID NOW is a Health Canada- and US Food and Drug Administration-approved isothermal nucleic acid amplification near-patient test that detects the RNA-dependent RNA polymerase gene and provides a qualitative result in 15 minutes. The assay is validated for use with nasal, nasopharyngeal, and throat swabs. The pooled clinical sensitivity and specificity of the ID NOW assay compared with RT-PCR are 73% [95% confidence interval (CI): 69%–76%] and 99.1% (95% CI: 99.0%–99.2%), respectively (21). As the device was not approved at the time of testing for use with buccal swabs, Health Canada approval was sought and provided prior to initiating the study.

## Buccal swab collection

The standard swab provided by the manufacturer was replaced with a Copan Standard Flocked swab [FLOQSwab (single-wrapped)] as these swabs have improved viral nucleic acid elution from saliva samples when compared with traditional swabs (22). Participants refrained from having food or drink for 10 minutes before buccal swab collection. Once ready, they were asked to pool saliva and secretions in their mouth. With the stick end of the sterile swab held between the thumb and forefingers, the swab was run along the inside of the mouth between the cheek and upper gum for approximately 20 seconds using an up-and-down and a back-forth motion. The applicator shaft was handed to the study team member who placed the swab into a sterile package. Testing was performed immediately by a research team member on the ABBOTT ID NOW devices located in the ED.

## Outcomes

The primary outcome was the detection of SARS-CoV-2. Secondary outcomes included characteristics associated with diagnostic accuracy, pain, and discomfort experienced with specimen sampling, and caregiver reported ease of buccal swab collection.

## Data collection

Data were collected via caregiver interviews conducted during the ED visit supplemented by a medical record review. Baseline symptoms were those present between symptom onset and the ED visit. Testing and reporting of variant of concern (VoC) varied by institution and over time. When a VoC or a mutation linked to a VoC was identified, that report was used for classification purposes. Race and ethnicity were self-reported by study participants or their caregivers.

After both swabs were collected, caregivers of participants <12.0 years of age were asked, using a 5-point Likert scale, to report (i) degree of pain and discomfort associated with swab performance and (ii) ease of buccal swab collection. Participants aged 5.0 to <18.0 years self-reported responses to these same questions. Higher scores indicated a greater amount of pain and discomfort. Participants aged 5.0 to <12.0 reported the level of pain experienced during the collection of the nasopharyngeal and buccal swabs using the Faces Pain Scale Revised Questionnaire with 0 being no pain and 10 being the worst pain (23). Those aged 12.0 to < 18.0 years reported the degree of pain and discomfort associated with both swabs using a 10-point verbal numeric scale where 0 = no pain and 10 = worst pain.

## Sample size

We determined *a priori* that enrolling 2,334 children would provide sufficient power to confirm a sensitivity of 80%, relative to the reference standard, assuming a prevalence of 3% with a confidence window of ±10%. Prevalence was based on ED-positivity rates at the time the study was designed, while the minimum sensitivity selected was based on the World Health Organization's lower limit of an "acceptable" sensitivity (24). After adjusting for 10% protocol non-adherence (e.g., inadequate or indeterminate specimens) a total of 2,882 participants was required.

## Statistical analysis

Data were summarized using descriptive statistics. Likert scale data are presented using frequency distributions (25). We classified participants as having concordant test results if both laboratory-based RT-PCR and the buccal swab ID NOW were the same and otherwise considered them discordant. To quantify agreement, we report percent agreement and used the Cohen's kappa (κ) statistic to assess the level of agreement between two tests beyond chance (26). We report sensitivity, specificity, negative and positive predictive values, and accuracy, along with 95% CIs, with the laboratory-based RT-PCR serving as the referent standard. Sub-group analyses based on age, buccal swab performer, vaccination status, and symptom duration, are reported. The Wilson score proportion differences test was used to obtain the 95% CI of the difference between proportions. Median cycle threshold (Ct) values, which refers to the number of cycles an RT-PCR assay needed to amplify viral RNA to reach detection (27), were compared between concordant and discordant participants using the Ct value of the positive standard-of-care swab and the Wilcoxon test, as an exploratory post hoc analysis.

To identify factors associated with diagnostic accuracy, a logistic regression model was fitted with the following covariates: site, sex, age, individual who administered the buccal swab (i.e., self, caregiver, or research assistant), vaccination status (i.e., unvaccinated, one dose, or ≥2 doses), and symptom duration. Sites with less than 40 observations were aggregated into a site level named "Smaller sites." Secondary outcomes of pain and discomfort were modeled using ordinal logistic regression (25). Statistical tests were two-sided, and statistical significance was defined as a $P < 0.05$. Analyses were performed with R, version 4.2.1.

## RESULTS

Of 3,838 children screened, 73.1% (2,804/3,838) provided consent, among whom 94.2% (2,640/2,804) met eligibility criteria (Fig. 1). Study participant median age was 3.0 years [interquartile range (IQR): 0.9 years, 8.0 years], 57.2% (1,509/2,640) were male, and 26.1% (690/2,640) received ≥1 COVID vaccine dose (Table 1). Although participants with discordant swab results more often reported a close SARS-CoV-2-positive contact, symptoms were similar between concordant and discordant groups. Fourteen percent (381/2,640) of participants were SARS-CoV-2 RT-PCR-positive and 41.2% (157/381) had VoC testing performed. Omicron was identified in 134 (85.4%) specimens; the remainder had unspecified lineages or were inconclusive.

## Diagnostic accuracy

Of the 2,640 specimens, 223 (8.4%) and 2,244 (85.0%) were concordant test-positive and concordant test-negative, respectively. A total of 158 (6.0%) participants tested positive on the nasopharyngeal RT-PCR swab but negative on the ID NOW buccal swab, while 15 (0.6%) participants tested positive on the ID NOW buccal swab and negative via nasopharyngeal RT-PCR (Table S1). Sensitivity and specificity of the ID NOW were 58.5% (95% CI: 53.4, 63.5) and 99.3% (95% CI: 98.9, 99.6), respectively; κ was 0.69 (95% CI: 0.64, 0.73) (Table 2). Positive and negative predictive values and overall accuracy were 93.7% (95% CI: 89.8, 96.4), 93.4% (95% CI: 92.4, 94.4), and 93.4% (95% CI: 92.4, 94.4), respectively.

One site was independently associated with test agreement (adjusted odds ratio: 3.84; 95% CI: 1.27, 11.6) (Table S2). Ct values were lower among concordant [median 17 (15, 21)] relative to discordant [median 30 (22, 35)] swabs ($P < 0.001$) (Fig. 2; Fig. S1).

## Sub-group analyses

Sensitivity was greater among children <4 years (68.2%; 95% CI: 62.2, 73.8) compared to 15 to <18 years (26.3%; 95% CI: 9.1, 51.2), with a difference of 41.9% (95% CI: 18.7, 57.3) (Table 2). Sensitivity varied according to swab performer, being greatest when the caregiver performed the swab (61.6%; 95% CI: 55.4, 67.6) and lowest when performed by the participant (45.3%; 95% CI: 32.8, 58.3); with a difference of 16.3% (95% CI: 2.8,

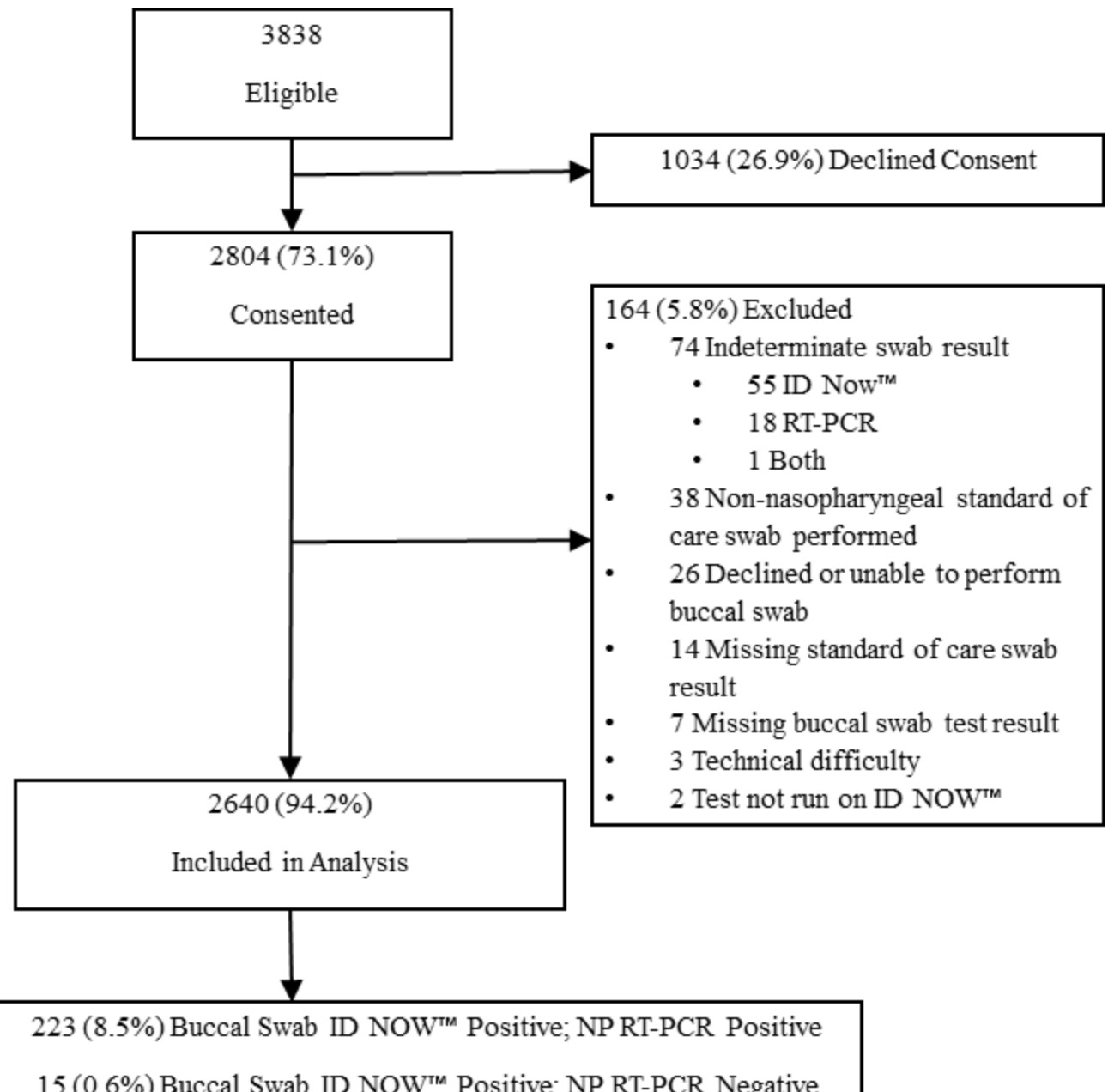

NP, nasopharyngeal; RT-PCR, reverse transcription polymerase chain reaction.

FIG 1   Flow diagram of study participants along with inclusions and exclusion criteria.

29.2). It also varied with vaccination status being 65.1% (95% CI: 59.5, 70.3) among unvaccinated participants and 25.0% (95% CI: 13.2, 40.3) among those who had received ≥2 doses; with a difference of 40.1% (95% CI: 24.7, 51.7). Lastly, sensitivity was greatest among those ill ≤2 days (65.3%; 95% CI: 59.1, 71.2) and lowest among those with >6 days of symptoms (35.4%; 95% CI: 22.2, 50.5); with a difference of 29.9% (95% CI: 14.5,

**TABLE 1** Participant demographic characteristics, epidemiologic risk factors, and behaviors and symptoms, reported as overall and based on concordant and discordant specimens

| Characteristic | All eligible participants 2,640 | Concordant results N = 2,467 | Discordant results N = 173 |
|---|---|---|---|
| Demographics | | | |
| Age, years, median (IQR) | 3.0 (0.9, 8.0) | 3.0 (0.9,8.0) | 3.0 (0.5,9.0) |
| Sex, male, n (%) | 1,509 (57.2) | 1,414 (57.3) | 95 (54.9) |
| Racial or ethnic identity, n (%)[a] | | | |
| Black | 213 (8.1) | 197 (8.0) | 16 (9.2) |
| East Asian | 116 (4.4) | 110 (4.5) | 6 (3.5) |
| Indigenous[b] | 63 (2.4) | 58 (2.4) | 5 (2.9) |
| Latin American | 91 (3.4) | 79 (3.2) | 12 (6.9) |
| Middle Eastern | 135 (5.1) | 122 (4.9) | 13 (7.5) |
| South Asian | 239 (9.1) | 222 (9.0) | 17 (9.8) |
| Southeast Asian | 126 (4.8) | 119 (4.8) | 7 (4.0) |
| White | 1,280 (48.5) | 1,217 (49.3) | 63 (36.4) |
| Multiracial | 342 (13.0) | 313 (12.7) | 29 (16.8) |
| Unknown | 35 (1.3) | 30 (1.2) | 5 (2.9) |
| Chronic underlying condition[c], n (%) | 653 (24.8) | 609 (24.7) | 44 (25.4) |
| Number of COVID vaccine doses received, n (%)[c] | | | |
| 0 | 1,936 (73.7) | 1,811 (73.8) | 125 (72.3) |
| 1 | 141 (5.4) | 126 (5.1) | 15 (8.7) |
| >2 | 549 (20.9) | 516 (21.0) | 33 (19.1) |
| Epidemiologic risk factors in the 14 days prior to testing | | | |
| Close contact with probable/confirmed SARS-CoV-2 case, n (%)[d,c] | 426 (16.2) | 357 (14.5) | 69 (39.9) |
| Attend any social gatherings with >10 people, n (%)[c] | 849 (32.2) | 790 (32.1) | 59 (34.1) |
| Attending in-person daycare or school, n (%)[c] | 1374 (52.1) | 1297 (52.7) | 77 (44.5) |
| Travel outside of the country, n (%)[c] | 358 (13.6) | 335 (13.6) | 23 (13.3) |
| Eat out at a restaurant, n (%)[c] | | | |
| Never | 1,858 (70.6) | 1,729 (70.3) | 129 (74.6) |
| Once a week | 593 (22.5) | 553 (22.5) | 40 (23.1) |
| More than once a week | 181 (6.9) | 177 (7.2) | 4 (2.3) |
| Take a bus, n (%)[c] | | | |
| Never | 2,228 (84.6) | 2,082 (84.6) | 146 (84.4) |
| Once a week | 109 (4.1) | 100 (4.1) | 9 (5.2) |
| More than once a week | 297 (11.3) | 279 (11.3) | 18 (10.4) |
| Take a train, n (%)[c] | | | |
| Never | 2,473 (93.9) | 2,315 (94.1) | 158 (91.3) |
| Once a week | 98 (3.7) | 89 (3.6) | 9 (5.2) |
| More than once a week | 63 (2.4) | 57 (2.3) | 6 (3.5) |
| Take a taxi or rideshare, n (%)[c] | | | |
| Never | 2,474 (94.0) | 2,313 (94.0) | 161 (93.1) |
| Once a week | 98 (3.7) | 88 (3.6) | 10 (5.8) |
| More than once a week | 61 (2.3) | 59 (2.4) | 2 (1.2) |
| Wear a mask when in public, n (%)[c] | | | |
| Never | 697 (26.5) | 652 (26.5) | 45 (26.0) |
| Sometimes | 719 (27.3) | 665 (27.0) | 54 (31.2) |
| Always | 1,219 (46.3) | 114 (46.5) | 74 (42.8) |
| Wash/disinfect hands after returning home, n (%)[c] | | | |
| Never | 754 (28.6) | 709 (28.8) | 45 (26.0) |
| Sometimes | 598 (22.7) | 548 (22.3) | 50 (28.9) |
| Always | 1,282 (48.7) | 1,204 (48.9) | 78 (45.1) |
| Symptoms | | | |
| Number of symptoms, median (IQR) | 8 (5, 10) | 8.0 (5.0, 10.0) | 7.0 (4.0, 11.0) |

(*Continued on next page*)

**TABLE 1** Participant demographic characteristics, epidemiologic risk factors, and behaviors and symptoms, reported as overall and based on concordant and discordant specimens (*Continued*)

| Characteristic | All eligible participants 2,640 | Concordant results *N* = 2,467 | Discordant results *N* = 173 |
|---|---|---|---|
| Duration of current illness, days, median (IQR)[c] | 3 (1, 6) | 3 (1, 6) | 2 (1, 5) |

[a]May select more than one.

[b]Includes First Nation, Métis, and Inuit.

[c]Close contact (*N* = 6), data missing for chronic underlying condition (*N* = 5), daycare/school attendance (*N* =5 ), duration of illness (*N* = 11), eat at a restaurant (*N* = 8), number of COVID vaccine doses received (*N* = 14), social gatherings (*N* = 5), take a bus (*N* = 6), take a taxi/rideshare (*N* = 7), take a train (*N* = 6), travel (*N* = 5), wash/disinfect hands before eating (*N* = 5), wash/disinfect hands when returning home (*N* = 6), wear a mask (*N* = 5).

[d]Close contact defined as sharing the same classroom environment, travelling together in any kind of conveyance, or living in the same household.

43.2). Among unvaccinated children <4 years of age, with symptoms ≤2 days, and swabs performed by a caregiver, sensitivity was 70.9% (95% CI: 63.3, 77.7).

## Acceptability

Among children <12 years of age, caregivers "agreed/strongly agreed" that their child experienced minimal pain more often with buccal swab sampling compared with nasopharyngeal swabs: 93.6% (2,098/2,241) vs 12.9% (289/2,236), with a difference of 80.7% (95% CI: 78.9, 82.3) (Table 3). Children aged 5 to <18 years of age "agreed/strongly agreed" that they experienced minimal pain in relation to the buccal swab [903/963 (93.8%)] compared to the nasopharyngeal swab [199/962 (20.7%)]; with a difference of 73.1% (95% CI: 69.9, 75.9). Additionally, 96.4% (856/888) of participants and 95.2% (2,107/2,213) of caregivers "agreed/strongly agreed that buccal swab sampling was easy.

On the verbal numeric scale, 97.0% (95% CI: 91.5, 99.0%) of participants 12 to <18 years rated the pain and discomfort as 0 or one for the buccal swab, while 6.0% (95% CI: 2.8, 12.5%) rated the standard-of-care swab (difference = 91.0%; 95% CI: 82.5, 94.7) similarly. On a scale where a score of 0 for performing the buccal swab is very easy and 10 is very difficult, 94.3% (95% CI: 91.5, 96.2) of respondents found the buccal swab very easy to perform. Lastly, 97.4% (95% CI: 95.8, 98.4) of participants 5 to <12 years reported a pain score lower than 5 for buccal swab performance, while only 28.9% (95% CI: 25.4,

**TABLE 2** Comparison of SARS-CoV-2 detection results from healthcare provider collected nasopharyngeal swabs tested by RT-PCR and caregiver or self-collected buccal swabs tested with the ID NOW including sub-group analyses

| Characteristic | Sensitivity (%) | Specificity (%) | Ppv (%) | Npv (%) | Accuracy (%) | Cohen's κ |
|---|---|---|---|---|---|---|
| Overall (*n* = 2,640) | 58.5 (53.4, 63.5) | 99.3 (98.9, 99.6) | 93.7 (89.8, 96.4) | 93.4 (92.4, 94.4) | 93.4 (92.4, 94.4) | 0.69 (0.64, 0.73) |
| Age | | | | | | |
| 0 to <4 years (*n* = 1,479) | 68.2 (62.2, 73.8) | 98.9 (98.2, 99.4) | 93.2 (88.6, 96.3) | 93.6 (92.1, 94.8) | 93.5 (92.1, 94.7) | 0.75 (0.70, 0.80) |
| 4 to <7 years (*n* = 394) | 43.2 (27.1, 60.5) | 99.7 (98.4, 100) | 94.1 (71.3, 99.9) | 94.4 (91.6, 96.5) | 94.4 (91.7, 96.5) | 0.57 (0.41, 0.73) |
| 7 to <9 years (*n* = 169) | 50.0 (24.7, 75.3) | 99.3 (96.4, 100) | 88.9 (51.8, 99.7) | 95.0 (90.4, 97.8) | 94.7 (90.1, 97.5) | 0.61 (0.39, 0.84) |
| 9 to <12 years (*n* = 214) | 35.7 (18.6, 55.9) | 100 (98.0, 100) | 100 (69.2, 100) | 91.2 (86.4, 94.7) | 91.6 (87.0, 94.9) | 0.49 (0.30, 0.68) |
| 12 to <15 years (*n* = 219) | 30.0 (11.9, 54.3) | 100 (98.2, 100) | 100 (54.1, 100) | 93.4 (89.2, 96.4) | 93.6 (89.5, 96.5) | 0.44 (0.20, 0.67) |
| 15 to <18 years (*n* = 165) | 26.3 (9.1, 51.2) | 100 (97.5, 100) | 100 (47.8, 100) | 91.2 (85.8, 95.1) | 91.5 (86.2, 95.3) | 0.39 (0.14, 0.63) |
| Swab performer | | | | | | |
| Participant (*n* = 533) | 45.3 (32.8, 58.3) | 100 (99.2, 100) | 100 (88.1, 100) | 93.1 (90.5, 95.1) | 93.4 (91.0, 95.4) | 0.59 (0.47, 0.71) |
| Caregiver (*n* = 1,599) | 61.6 (55.4, 67.6) | 99.0 (98.3, 99.4) | 91.9 (86.8, 95.5) | 93.1 (91.6, 94.3) | 92.9 (91.6, 94.1) | 0.70 (0.65, 0.75) |
| Study team member (*n* = 508) | 59.3 (45.7, 71.9) | 99.8 (98.8, 100) | 97.2 (85.5, 99.9) | 94.9 (92.5, 96.7) | 95.1 (92.8, 96.8) | 0.71 (0.61, 0.82) |
| Vaccination status | | | | | | |
| Unvaccinated (*n* = 1,936) | 65.1 (59.5, 70.3) | 99.1 (98.5, 99.5) | 93.2 (89.0, 96.1) | 93.6 (92.3, 94.7) | 93.5 (92.4, 94.6) | 0.73 (0.69, 0.77) |
| One dose (*n* = 141) | 31.8 (13.9, 54.9) | 100 (96.9, 100) | 100 (59.0, 100) | 88.8 (82.2, 93.6) | 89.4 (83.1, 93.9) | 0.44 (0.22, 0.66) |
| More than two doses (*n* = 549) | 25.0 (13.2, 40.3) | 100 (99.3, 100) | 100 (71.5, 100) | 93.9 (91.5, 95.7) | 94.0 (91.7, 95.8) | 0.38 (0.22, 0.54) |
| Symptom duration | | | | | | |
| 0–2 days (*n* = 1,140) | 65.3 (59.1, 71.2) | 99.1 (98.2, 99.6) | 95.3 (91.0, 98.0) | 91.0 (89.0, 92.7) | 91.7 (89.9, 93.2) | 0.73 (0.68, 0.78) |
| 3–4 days (*n* = 568) | 50.9 (37.3, 64.4) | 99.4 (98.3, 99.9) | 90.6 (75.0, 98.0) | 94.8 (92.5, 96.5) | 94.5 (92.3, 96.3) | 0.62 (0.50, 0.74) |
| 5–6 days (*n* = 343) | 54.2 (32.8, 74.4) | 99.7 (98.3, 100) | 92.9 (66.1, 99.8) | 96.7 (94.1, 98.3) | 96.5 (94.0, 98.2) | 0.67 (0.49, 0.84) |
| > 6 days (*n* = 578) | 35.4 (22.2, 50.5) | 99.4 (98.4, 99.9) | 85.0 (62.1, 96.8) | 94.4 (92.2, 96.2) | 94.1 (91.9, 95.9) | 0.47 (0.33, 0.62) |

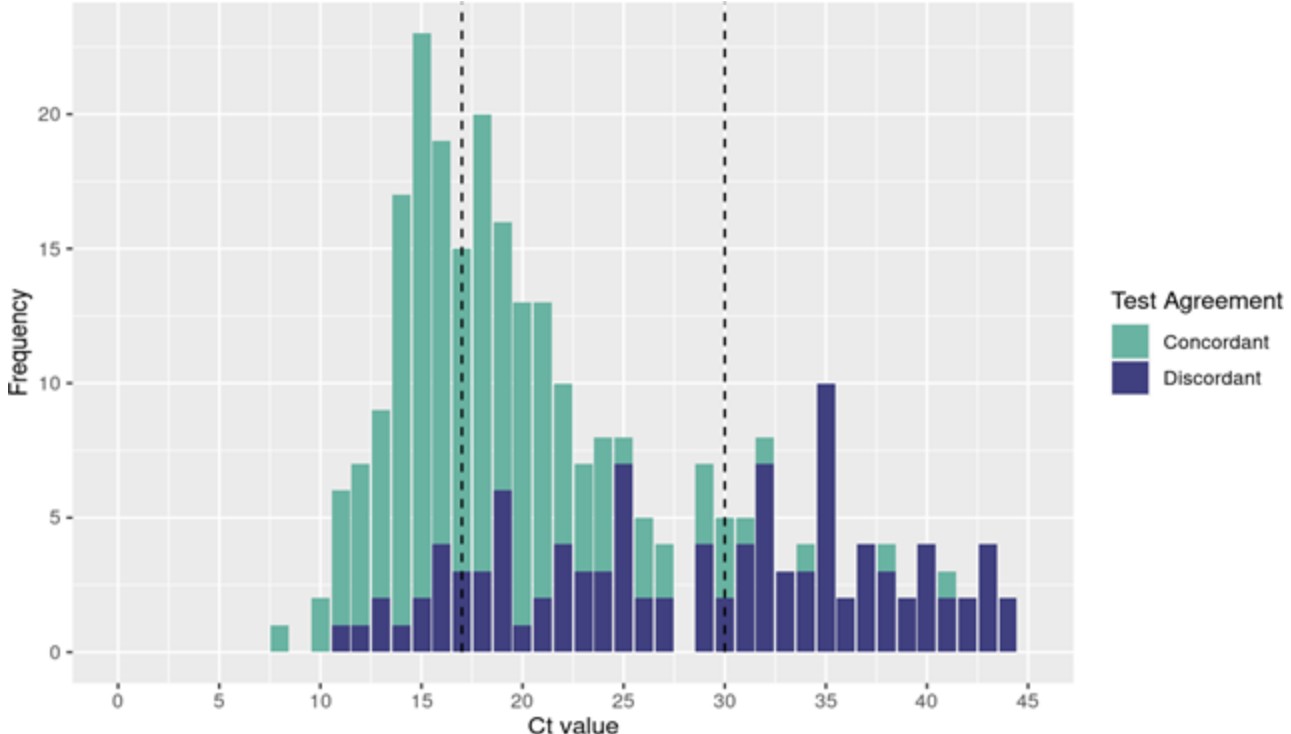

**FIG 2** Ct values of concordant (turquoise) and discordant (blue) swaps. Median values for the 273 patients with Ct values (vertical dashed lines) available differed between groups [concordant: 17(15, 21), discordant: 30(22, 35); P<0.001].

32.7) reported the standard-of-care swab as causing pain less than 5, with a difference of 68.5% (95% CI: 64.3, 72.2) (Fig. S2).

## DISCUSSION

In 14 Canadian pediatric EDs, we compared healthcare provider-collected nasopharyngeal specimens tested for SARS-CoV-2 via laboratory-based RT-PCR, with caregiver- or self-collected buccal swab specimen tested on a point-of-care nucleic acid amplification device and found that overall accuracy was 93%. Although specificity exceeded 99%, sensitivity was only 58.5%. Concordance was greater with higher viral loads. Children and their caregivers reported that buccal swabs cause less pain and discomfort than nasopharyngeal swabs and are easier to perform. Sensitivity was greatest among children <4 years of age (vs older children), when caregivers performed the swabs (vs self-swab), among unvaccinated children (vs vaccinated), and among those with a shorter duration of symptoms.

Previous studies of the ID NOW reported higher sensitivity. In a study comparing 12,821 nasal specimens tested on the ID NOW with combined nose and throat swabs tested on the Hologic Aptima transcription-mediated amplification device, sensitivity was 85% (28). An upgraded version of the ABBOTT ID NOW (version 2.0), was reported to have 98% (nasopharyngeal and oropharyngeal swabs)(29) and 95% (nasopharyngeal) sensitivity (30) with false-negative specimens having Ct values >35, reflecting very low viral loads (29). Our overall lower sensitivity was possibly due to our use of self-collected buccal swabs as we did find that sensitivity was greater when collection was performed by caregivers. Others have shown that healthcare worker-collected specimens (nose and throat) are more sensitive than self-collected specimens (31), and an adult study demonstrated that buccal swab SARS-CoV-2 viral RNA loads are lower after food intake (32), thus providing another avenue to increase buccal swab specimen sensitivity.

This study was performed almost entirely in the Omicron-dominant era, and in fact, Omicron was identified in 85% of specimens that had VoC testing performed with the

**TABLE 3** Participant and caregiver reported pain associated with buccal and nasopharyngeal swab performance and ease of buccal swab performance

| Participants <12 years – caregiver report | | |
|---|---|---|
| Swab performance was associated with minimal pain and discomfort for my child | | |
| | Buccal swab | Nasopharyngeal swab | P value |
| Strongly agree | 1,534 (68.5%) | 103 (4.6%) | <0.001 |
| Agree | 564 (25.2%) | 186 (8.3%) | |
| Neither agree nor disagree | 64 (2.9%) | 154 (6.9%) | |
| Disagree | 52 (2.3%) | 689 (30.8%) | |
| Strongly disagree | 27 (1.2%) | 1,104 (49.4%) | |
| Participants 5 to <18 years – participant report | | |
| Swab performance was associated with minimal pain and discomfort | | |
| | Buccal swab | Nasopharyngeal swab | P value |
| Strongly agree | 748 (77.7%) | 55 (5.7%) | <0.001 |
| Agree | 155 (16.1%) | 144 (15.0%) | |
| Neither agree nor disagree | 10 (1.0%) | 99 (10.3%) | |
| Disagree | 20 (2.1%) | 331 (34.4%) | |
| Strongly disagree | 30 (3.1%) | 333 (34.6%) | |
| Performing the cheek swab was easy[a] | | |
| | Participant 5 to <18 years of age | Caregiver <12 years of age | P value |
| Strongly agree | 723 (81.4%) | 1,654 (74.7%) | <0.001 |
| Agree | 133 (15.0%) | 453 (20.5%) | |
| Neither agree nor disagree | 25 (2.8%) | 59 (2.7%) | |
| Disagree | 5 (0.6%) | 34 (1.5%) | |
| Strongly disagree | 2 (0.2%) | 13 (0.6%) | |

[a]'**Performing the cheek swab was easy**': In the left column, the perception of the participant (if aged 5 to <18 years) is reported, and in the right column, the perception of the caregiver (if the participant was aged <12 years) is reported. The P value compares the ratings between these two groups.

remainder having unspecified lineages or were inconclusive. It should be noted that, due to mutation within the S gene, Omicron variants can have S gene target failure with some commercially available PCR kits. However, as the ID NOW assay targets the RdRp gene, this should not have affected assay performance. In concordance with site (Table S2), it is possible that standard-of-care test performance characteristics could have influenced our results in relation to the ID NOW. Lastly, it is unknown how buccal swab specimens would perform on the improved ID NOW or other more sensitive devices.

As SARS-CoV-2 infection in otherwise healthy children evaluated outside of the hospital setting is rarely associated with severe outcomes (33), the main reason for SARS-CoV-2 testing in children is to identify those likely to transmit infection to more vulnerable individuals. However, transmissibility is more complicated than the dichotomization provided by laboratory-based RT-PCR results (i.e., positive vs negative). Ct values, representing the number of amplification cycles required to detect viral genetic material, are inversely related to the amount of SARS-CoV-2 nucleic acid in the tested sample (i.e., when Ct values are high, viral load is low) (34). As such, lower Ct values are more strongly associated with infectiousness (35–37). With the majority of false negatives in our study having high Ct values, these likely were collected from individuals with non-transmissible nucleic acid. This is based on evidence that SARS-CoV-2 cannot be grown in cell culture from patients with high Ct values, with maximal values ranging from 24 (38) to 34 (39).

Among school-age children, parents, and school personnel who agreed to perform COVID-19 surveillance via a saliva sampling method, 36% indicated they would refuse testing had a nasal swab approach been required (40). When stratified by school level, 51% of elementary school students or their caregivers would not agree to testing if a nasal swab was offered. This reluctance to be tested in the pediatric population poses problems for infection control during pandemics. While nasopharyngeal swab acceptability ranged from 79% to 100% in a cross-sectional survey administered to children

6–17 years of age and their caregivers, saliva sampling acceptability was consistently higher, ranging from 92% to 98% (41). While 58% of youth describe significant pain with nasopharyngeal swabbing, none report significant pain with saliva sampling and 90% of children state they prefer saliva sampling if given a choice. Thus, alternatives to nasopharyngeal swabs are needed. We evaluated buccal swabs as they result in less discomfort, can be performed by non-healthcare personnel, and have fewer contraindications (e.g., nasal anatomic pathology, and coagulopathy) and complications (e.g., retained swabs, epistaxis, and cerebrospinal fluid leak) (42).

At this time, nasopharyngeal swabs remain the standard-of-care, particularly in the hospital setting, even though nasal swab collection and antigen testing have acceptable sensitivity (~80%) (43). Our results do not support the use of buccal swab ID NOW testing at this time in such settings. While the greater acceptability of buccal swabs is a crucial finding, methods to improve sensitivity are required and could include use of a different device to detect SARS-CoV-2, or performing testing using laboratory-based RT-PCR. The COVID-19 pandemic has demonstrated how need brings innovation; thus, in the future, cost-effective, easy-to-operate, point-of-care devices, with higher sensitivity will likely be available.

A strength of our study is the multi-institutional design that allowed the buccal swab ID NOW approach to be compared across a variety of populations of varying acuity and viral load scenarios. This is also a limitation, as each site had its own clinically validated standard-of-care assay with different performance characteristics (i.e., sensitivity, specificity, and Ct value positivity cut point) and criteria and approach to performing VoC testing. In addition, Ct values are imperfect representations of viral load, varying significantly between platforms and between gene targets within test systems (44). Nonetheless, as some laboratory assays did not produce Ct values, we were unable to obtain Ct results on all laboratory-based SARS-CoV-2-positive RT-PCR specimens, limiting our ability to interpret all false-negative results. Moreover, the buccal swab specimens were run on a qualitative device, and as such, Ct values are unavailable for those specimens. Research team members at some sites did perform a small number of the buccal swabs. The exclusion of children who failed to perform the buccal swab may have affected our results related to the pain and ease of sampling. However, this represented only 1% of the study population. A single SARS-CoV-2 RT-PCR is an imperfect reference standard (45); children with positive ID NOW tests with negative RT-PCR tests may be true positives, and the specificity thus may be underestimated. On the other hand, sensitivity may have been underestimated as we utilized the reference laboratory's RT-PCR result interpretation as the gold standard, even when exceedingly high Ct values were reported. Lastly, to optimize sensitivity, we replaced the swab provided by the manufacturer by a flocked swab, as such our results cannot be generalized to the device and swab as provided by the manufacturer.

In conclusion, although overall accuracy of the buccal swab-point-of-care SARS-CoV-2 test compared with nasopharyngeal swab laboratory-based RT-PCR was good and the approach involving buccal swabs was much less painful, sensitivity was too low to permit adoption of this approach. However, sensitivity was greatest among children <4 years of age (vs older children), when caregivers performed the swabs (vs self-swab), among unvaccinated children (vs. vaccinated), and among those with a shorter duration of symptoms. While negative agreement was excellent, positive percent agreement was <60%. Concordance was greater among those with higher viral loads.

## ACKNOWLEDGMENTS

Support for this study was provided by the Health Canada's Safe Restart Agreement Contribution Program. S.F. is supported by the Alberta Children's Hospital Foundation Professorship in Child Health and Wellness.

None of the funders played any role in the design or conduct of the study, collection, management, analysis, or interpretation of the data, or in the preparation, review, or

approval of the manuscript and decision to submit the manuscript for publication. The views expressed herein do not necessarily represent the views of Health Canada.

The PERC COVID Study Group Non-Author Contributors:

Samina Ali, Waleed Alqurashi, Tyrus Crawford, Rebecca Emerton, Cathie-Kim Le, Melissa Lorenzo, Candice McGahern, Garth Meckler, and Mandi Newton.

## AUTHOR AFFILIATIONS

[1]Department of Pediatrics, University of Calgary, Calgary, Alberta, Canada

[2]Department of Emergency Medicine, University of Calgary, Calgary, Alberta, Canada

[3]Sections of Pediatric Emergency Medicine and Gastroenterology, University of Calgary, Calgary, Alberta, Canada

[4]Cumming School of Medicine, University of Calgary, Calgary, Alberta, Canada

[5]Department of Pediatrics, University of Calgary, Calgary, Alberta, Canada

[6]Cumming School of Medicine, University of Calgary, Calgary, Alberta, Canada

[7]Division of Emergency Medicine, Department of Paediatrics, Hospital for Sick Children, Faculty of Medicine, University of Toronto, Toronto, Ontario, Canada

[8]Division of Emergency Medicine, McMaster Children's Hospital, Hamilton, Ontario, Canada

[9]Department of Pediatrics, McMaster Children's Hospital, Hamilton, Ontario, Canada

[10]Department of Paediatrics, BC Children's Hospital Research Institute, University of British Columbia, Vancouver, British Columbia, Canada

[11]Department of Emergency Medicine, BC Children's Hospital Research Institute, University of British Columbia, Vancouver, British Columbia, Canada

[12]Division of Pediatric Emergency Medicine, Department of Pediatrics, University of Alberta, Edmonton, Canada

[13]Department of Pediatrics, Women's and Children's Health Research Institute, University of Alberta, Edmonton, Canada

[14]Department of Pediatrics, University of Ottawa, Children's Hospital of Eastern Ontario, Ottawa, Ontario, Canada

[15]Département de médecine de famille et de médecine d'urgence, CHU de Québec-Université Laval, Québec City, Quebec, Canada

[16]Department of Pediatric Emergency Medicine, Centre Hospitalier Universitaire (CHU) Sainte-Justine, Université de Montréal, Montreal, Quebec, Canada

[17]Division of Pediatric Emergency Medicine, Montreal Children's Hospital, McGill University Health Centre, Montreal, Quebec, Canada

[18]Department of Pediatrics, Montreal Children's Hospital, McGill University Health Centre, Montreal, Quebec, Canada

[19]Department of Epidemiology, Biostatistics, and Occupational Health, McGill University, Montreal, Quebec, Canada

[20]Department of Emergency Medicine, IWK Children's Health Centre, Queen Elizabeth II Health Sciences Centre, Dalhousie University, Halifax, Nova Scotia, Canada

[21]Section of Pediatric Emergency, Jim Pattison Children's Hospital, University of Saskatchewan, Saskatoon, Saskatchewan, Canada

[22]Department of Pediatrics, Jim Pattison Children's Hospital, University of Saskatchewan, Saskatoon, Saskatchewan, Canada

[23]Janeway Children's Health and Rehabilitation Centre, NL Health Services, St. John's, Newfoundland and Labrador, Canada

[24]Department of Paediatrics, Children's Hospital London Health Sciences Centre, Schulich School of Medicine and Dentistry, London, Ontario, Canada

[25]Department of Internal Medicine, Schulich School of Medicine and Dentistry, London, Ontario, Canada

[26]Department of Epidemiology and Biostatistics, Schulich School of Medicine and Dentistry, London, Ontario, Canada

[27]Clinical Research Unit, Children's Hospital of Eastern Ontario Research Institute, Ottawa, Ontario, Canada

28Department of Pathology and Laboratory Medicine, BC Children's Hospital Research Institute, University of British Columbia, Vancouver, British Columbia, Canada

29Department of Paediatrics, Kingston Health Sciences Centre, Queen's University, Kingston, Ontario, Canada

30Department of Emergency Medicine, University of Ottawa, Children's Hospital of Eastern Ontario, Ottawa, Ontario, Canada

## AUTHOR ORCIDs

Stephen B. Freedman http://orcid.org/0000-0003-2319-6192
Bruce Wright http://orcid.org/0000-0002-5056-9356
Maala Bhatt http://orcid.org/0000-0003-1265-3482
Jocelyn Gravel http://orcid.org/0000-0001-5901-4990
Brett Burstein http://orcid.org/0000-0002-1107-9514
Ahmed Mater http://orcid.org/0000-0003-2319-6192
Robert Porter http://orcid.org/0000-0001-5505-0716
Deepti Reddy http://orcid.org/0000-0003-2319-6192
Richard J. Webster http://orcid.org/0000-0003-2319-6192
David M. Goldfarb http://orcid.org/0000-0003-0835-9504

## FUNDING

| Funder | Grant(s) | Author(s) |
|---|---|---|
| Canadian Government \| Health Canada (Santé Canada) | Safe Restart Agreement Contribution Program | Stephen B. Freedman |
| Alberta Children's Hospital Foundation | Professorship in Child Health and Wellness | Stephen B. Freedman |

## AUTHOR CONTRIBUTIONS

Stephen B. Freedman, Conceptualization, Data curation, Formal analysis, Funding acquisition, Investigation, Methodology, Project administration, Resources, Software, Supervision, Writing – original draft | Kelly Kim, Conceptualization, Data curation, Funding acquisition, Investigation, Methodology, Project administration, Resources, Writing – review and editing | Gabrielle Freire, Investigation, Methodology, Project administration, Supervision, Writing – review and editing | Alicia Kanngiesser, Data curation, Investigation, Methodology, Writing – review and editing | April Kam, Investigation, Methodology, Project administration, Supervision, Writing – review and editing | Quynh Doan, Investigation, Methodology, Project administration, Supervision, Writing – review and editing | Bruce Wright, Investigation, Methodology, Project administration, Supervision, Writing – review and editing | Maala Bhatt, Investigation, Methodology, Project administration, Supervision, Writing – review and editing | Simon Berthelot, Investigation, Methodology, Project administration, Supervision, Writing – review and editing | Jocelyn Gravel, Investigation, Methodology, Project administration, Supervision, Writing – review and editing | Brett Burstein, Investigation, Methodology, Project administration, Supervision, Writing – review and editing | Jason Emsley, Investigation, Methodology, Project administration, Supervision, Writing – review and editing | Ahmed Mater, Investigation, Methodology, Project administration, Supervision, Writing – review and editing | Robert Porter, Investigation, Methodology, Project administration, Supervision, Writing – review and editing | Naveen Poonai, Investigation, Methodology, Project administration, Supervision, Writing – review and editing | Deepti Reddy, Formal analysis, Methodology, Visualization, Writing – review and editing | Richard J. Webster, Formal analysis, Funding acquisition, Methodology, Resources, Software, Supervision, Writing – review and editing | David M. Goldfarb, Conceptualization, Funding acquisition, Investigation, Methodology, Writing – review and editing | Kirk Leifso, Investigation, Methodology, Project administration, Supervision, Writing –

review and editing | Roger Zemek, Conceptualization, Funding acquisition, Investigation, Methodology, Project administration, Resources, Software, Supervision, Writing – review and editing.

## DATA AVAILABILITY

Data will be made available upon request in accordance with privacy legislation, consent provided, appropriate data sharing agreements are established, and appropriate ethical approvals are obtained.

## ADDITIONAL FILES

The following material is available online.

### Supplemental Material

**Supplemental tables and figures (Spectrum01884-24-s0001.docx).** Tables S1 and S2; Fig. S1 and S2.

### Open Peer Review

**PEER REVIEW HISTORY (review-history.pdf).** An accounting of the reviewer comments and feedback.

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
