## [Reviewer comments · Microbiology Spectrum]

Microbiology Spectrum

Accuracy of Point-of-Care SARS-CoV-2 Detection Using Buccal Swabs in Pediatric Emergency Departments

Stephen Freedman, Kelly Kim, Gabrielle Friere, Alicia Kanngiesser, April Kam, Quynh Doan, Bruce Wright, Maala Bhatt, Simon Berthelot, Jocelyn Gravel, Brett Burstein, Jason Emsley, Ahmed Mater, Robert Porter, Naveen Poonai, Deepti Reddy, Richard Webster, David Goldfarb, Kirk Leifso, and Roger Zemek

Corresponding Author(s): Stephen Freedman, University of Calgary

Review Timeline:

Submission Date:	July 28, 2024
Editorial Decision:	August 29, 2024
Revision Received:	September 6, 2024
Accepted:	September 17, 2024

Editor: Eleanor Powell

Reviewer(s): Disclosure of reviewer identity is with reference to reviewer comments included in decision letter(s). The following individuals involved in review of your submission have agreed to reveal their identity: James J Dunn (Reviewer #2)

Transaction Report:

DOI: <https://doi.org/10.1128/spectrum.01884-24>

Re: Spectrum01884-24 (Accuracy of Point-of-Care SARS-CoV-2 Detection Using Buccal Swabs in Pediatric Emergency Departments)

Dear Dr. Stephen Freedman:

Thank you for the privilege of reviewing your work. After receiving feedback from two reviewers, modifications are required before potential publication. Below you will find instructions from the Spectrum editorial office and the reviewer comments.

Revision Guidelines

Sincerely,
Eleanor Powell
Editor
Microbiology Spectrum

Reviewer #1 (Comments for the Author):

The authors present a multicenter study evaluating the performance of buccal swabs using a flocked swab and the ID NOW assay. Although the specificity was high, the sensitivity was low despite the use of a higher quality swab.

1. The authors have framed the manuscript to recommend when a buccal swab might be acceptable. However, given the very

low sensitivity (<60%), I disagree with that approach. Rather, it should be presented as a well-performed negative data study. Over-the-counter nasal swab antigen tests are available with a significantly higher sensitivity than buccal swabs. Why would there be a recommendation to use a test with such poor performance in a hospital setting when there are much better options?

2. Despite my disagreement with the conclusions, the study is thorough, very well-performed, and the manuscript well written.

3. Line 301 has a typo.

Reviewer #2 (Comments for the Author):

The study by Freedman et al. examined the performance characteristics and satisfaction of provider or self-collection of buccal swabs for SARS-CoV-2 detection on the Abbott ID NOW at the point of care compared to the reference standard of NP swab collection by a provider and virus detection in the clinical lab using RT-PCR. The authors found that although the buccal swab was easier to collect and better tolerated, the sensitivity of SARS-CoV-2 detection was only 58.5% on the ID NOW compared to an NP swab tested by RT-PCR. The paper is clearly and concisely written and generally the appropriate statistics have been applied.

I think some information is needed on the comparator RT-PCR or other nucleic acid amplification assays used for SOC testing. Each of these may have different performance characteristics themselves (i.e. sensitivity, specificity). In addition, it is not always appropriate to equate the Ct values from one platform/assay to that of another.

It would be worthwhile to know more about the clinical characteristics of the 15 patients that were ID NOW-positive and RT-PCR-negative. What age group, symptoms and duration, known exposures, vaccination status, etc. Perhaps this can be detailed in a supplementary table and mentioned in the discussion.

Lines 202-204: The methods for VoC determination are not described. Perhaps a reference could be added here. I think it is worthwhile to know the predominant strain during this time period (it appears to be omicron in this study) and it would also be important to know if there were variants that might have affected the performance of the RT-PCR assays, particularly the negatives that were ID NOW-positive. During that time, assay performance was predominantly affected by S gene target failure.

Line 259: Specify the characteristics refer to buccal swab testing.

Line 264: It appears from eTable 1 that there two sites more likely to have test agreement. That could be influenced by the performance characteristics of the SOC test relative to ID NOW. Discussion of this should be included.

Table 3: Under "Performing cheek swab was easy" it's not clear which data the p value statistical comparison applies to. The two data columns listed here appear to be statistically similar. Should the statistically significant difference refer to the ease of buccal vs. NP swab collection? The ease of NP swab collection data is not shown.

Figure 1: For the 74 indeterminate swabs excluded please separate these as ID NOW indeterminate and RT-PCR (or SOC test) indeterminate. It would be important to know with regard to platform performance.

Reference 21: This title should include "molecular-based tests" based on PMID 33760236.

**DEPARTMENT OF PEDIATRICS
ALBERTA CHILDREN'S HOSPITAL
28 Oki Drive NW
Calgary, AB Canada T3B 6A8**

September 5, 2024

Manuscript #: Spectrum01884-24

Title: Accuracy of Point-of-Care SARS-CoV-2 Detection Using Buccal Swabs in Pediatric Emergency Departments

Dear Dr. Powell,

Thank you for providing us the opportunity to address the comments provided by your Reviewers. We have carefully considered all comments and requirements and have revised our manuscript accordingly whenever possible. We address each comment on a point-by-point basis below. We hope that our responses are acceptable, and we would be happy to discuss any outstanding concerns.

Thank you for reviewing our manuscript.

Sincerely,

Stephen Freedman, MDCM, MSc, FAAP, FRCPC
Sections of Pediatric Emergency Medicine and Gastroenterology
Alberta Children's Hospital
Alberta Children's Hospital Foundation, Professor in Child Health and Wellness
University of Calgary

Comment	Response	Changes/Location
Reviewer #1		
The authors have framed the manuscript to recommend when a buccal swab might be acceptable. However, given the very low sensitivity (<60%), I disagree with that approach. Rather, it should be presented as a well-performed negative data study. Over-the-counter nasal swab antigen tests are available with a significantly higher sensitivity than buccal swabs. Why would there be a recommendation to use a test with such poor performance in a hospital setting when there are much better options?	We appreciate hearing this feedback and have revised the discussion to reflect this perspective by the removal of certain sentences from the discussion and a revision to the Concluding paragraph.	Page 5: Abstract: “Conclusions: Although accuracy of the buccal swab point-of-care SARS-CoV-2 test was good, and negative agreement was excellent, sensitivity was only 58.5%.” Page 19: Discussion: “‘At this time, nasopharyngeal swabs remain the standard-of-care, particularly in the hospital setting, even though nasal swab collection and antigen testing have acceptable sensitivity (~80%).(43) Our results do not support the use of buccal swab–ID NOW™ testing at this time in such settings. While the greater acceptability of buccal swabs is a crucial finding, methods to improve sensitivity are required and could include use of a different device to detect SARS-CoV-2, or performing testing using laboratory-based RT-PCR.” Page 21: “In conclusion, although overall accuracy of the buccal swab-point-of-care SARS-CoV-2 test compared with nasopharyngeal swab laboratory-based RT-PCR was good and the approach involving buccal swabs was much less painful, sensitivity was too low to permit adoption of this approach.”
Line 301 has a typo.	Revised – thank you.	
Reviewer #2		

Comment	Response	Changes/Location
I think some information is needed on the comparator RT-PCR or other nucleic acid amplification assays used for SOC testing. Each of these may have different performance characteristics themselves (i.e. sensitivity, specificity).	Unfortunately gathering this information is not feasible as all sites had their own approach and this approach changed over time and given the time interval that has passed it is not possible to track down all tests performed and the dates of performance and the many changes that occurred at all sites during the study period. However, as all laboratories employed clinically validated tests, we do not think this was likely to have a meaningful impact on our results and interpretation. We have added test to the Discussion, Limitations section to address this point.	Page 20: Discussion: “A strength of our study is the multi-institutional design that allowed the buccal swab–ID NOW™ approach to be compared across a variety of populations of varying acuity and viral load scenarios. This is also a limitation, as each site had its own clinically validated standard of care assay with different performance characteristics (i.e., sensitivity, specificity and Ct value positivity cut point), and criteria and approach to performing VoC testing. In addition, Ct values are imperfect representations of viral load, varying significantly between platforms and between gene targets within test systems.(44)”
In addition, it is not always appropriate to equate the Ct values from one platform/assay to that of another.	We concur with this comment and have added it as a limitation.	Page 20: Discussion: “In addition, Ct values are imperfect representations of viral load, varying significantly between platforms and between gene targets within test systems (44).”
It would be worthwhile to know more about the clinical characteristics of the 15 patients that were ID NOW-positive and RT-PCR-negative. What age group, symptoms and duration, known exposures, vaccination status, etc. Perhaps this can be detailed in a supplementary table and mentioned in the discussion.	We appreciate this suggestion and have added a new table to the supplement that provides the information requested (eTable 1).	See eTable 1.
Lines 202-204: The methods for VoC determination are not described. Perhaps a reference could be added here. I think	We appreciate this comment and have addressed it in several ways. 1) Since performance of VoC testing	1) Page 20: Discussion: “A strength of our study is the multi-institutional design that allowed the buccal swab–

Comment	Response	Changes/Location
it is worthwhile to know the predominant strain during this time period (it appears to be omicron in this study) and it would also be important to know if there were variants that might have affected the performance of the RT-PCR assays, particularly the negatives that were ID NOW-positive. During that time, assay performance was predominantly affected by S gene target failure.	and diagnostic approach varied across sites and over time we cannot succinctly describe how it was performed in our methods. We have added this as a limitation. 2) We have provided results of VoC testing that do confirm that Omicron was indeed the only variant definitively identified and that testing was performed almost exclusively in the Omicron dominant era. 3) We have added a sentence to the Discussion regarding the potential impact of S gene target failure. 4) We unfortunately do not have VoC data related to ID NOW positive, lab RT-PCR negative specimens as these were not sequenced by the lab given that they tested negative by RT-PCR.	ID NOW™ approach to be compared across a variety of populations of varying acuity and viral load scenarios. This is also a limitation, as each site had its own clinically validated standard of care assay with different performance characteristics (i.e., sensitivity, specificity and Ct value positivity cut point), and criteria and approach to performing VoC testing.” 2) Page 14: Results: “Omicron was identified in 134 (85.4%) specimens; the remainder had unspecified lineages or were inconclusive.” Page 18: Discussion: “This study was performed almost entirely in the Omicron dominant era, and in fact Omicron was identified in 85% of specimens that had VoC testing performed with the remainder having unspecified lineages or were inconclusive.” 3) Page 18: Discussion: “It should be noted, that due to mutation within the S gene, Omicron variants can have S gene target failure with some commercially available PCR kits. However, as the ID NOW™ assay targets the RdRp gene, this should not have affected assay performance.”
Line 259: Specify the characteristics	Revised as suggested.	Page 14: Results: “Sensitivity and specificity

Comment	Response	Changes/Location
refer to buccal swab testing.		of the ID NOW™ were 58.5% (95%CI: 53.4, 63.5) and 99.3% (95%CI: 98.9, 99.6), respectively; kappa (κ) was 0.69 (95%CI: 0.64, 0.73); Table 2. ”
Line 264: It appears from eTable 1 that there two sites more likely to have test agreement. That could be influenced by the performance characteristics of the SOC test relative to ID NOW. Discussion of this should be included.	We appreciate this comment and have added a sentence to the Discussion on this topic.	Page 18: Discussion: “As concordance with associated with site (eTable 2) it is possible that standard of care test performance characteristics could have influenced our results in relation to the ID NOW™.”
Table 3: Under "Performing cheek swab was easy" it's not clear which data the p value statistical comparison applies to. The two data columns listed here appear to be statistically similar. Should the statistically significant difference refer to the ease of buccal vs. NP swab collection? The ease of NP swab collection data is not shown.	This table is in fact depicting what the Reviewer is asking about. Under the header ‘Performing the Cheek Swab was Easy’ we report in the left column the perception of the participant (if aged 5 to < 18 years) and in the right column the perception of the caregiver (if the participant was aged < 12 years). The P value compares the ratings between these two groups and reveals that participants report that performing the cheek swab was easier than reported by caregivers. We could not compare this to NP swab as those were performed by nurses. Our biostatisticians have reviewed the results of the analysis and after auditing our code, they have confirmed that the reported P value is correct based on the results of our planned ordinal logistic regression analysis. There are two reasons why it may be hard to ‘see the	Page 31: Table 3 Footnote: “*‘ Performing the Cheek Swab was Easy ’: In the left column the perception of the participant (if aged 5 to < 18 years) is reported and in the right column the perception of the caregiver (if the participant was aged < 12 years) is reported. The P value compares the ratings between these two groups.”

Comment	Response	Changes/Location
	difference’:  1) The difference is being driven by the ‘strongly agree’ level (as most of the data is there). As we have a large sample in the ‘strongly agree’ level, a modest difference will be statistically significant. To confirm our results, we performed an alternative descriptive comparison, using the Wilson’s score difference between proportions. Reassuringly this found that participants were 6.7% (95% CI: 3.5 - 9.7%) more likely to report ‘strongly agree’ compared to caregivers; this difference between participants and caregivers’ proportions of ‘strongly agree’ has non-overlapping 95%CI. 2) The eye-ball test to compare differences at each level from the table (e.g., within just ‘strongly agree’, within just ‘agree’, etc.), does not match how the ordinal regression generates our single p-value. Ordinal logistic regression is looking to understand how much closer each predictor (i.e., 	

Comment	Response	Changes/Location
	participant vs. caregiver) pushes the outcome towards an ordinal increase for the outcome (e.g., a jump up on the Likert scale). This difference may not be evident when looking at the raw proportions but becomes apparent when considering the cumulative proportions across all levels of agreement. Consequently, the differences for the ‘strongly agree’ level persists when we perform an ordinal analysis (e.g., ‘strongly agree’, vs others; ‘strongly agree’ + ‘agree’ vs all others, etc.). Similarly, as most of the data is in the ‘strongly agree’ level, this dominates the single p-value provided by the ordinal regression, irrespective of pairwise comparisons.	
Figure 1: For the 74 indeterminate swabs excluded please separate these as ID NOW indeterminate and RT-PCR (or SOC test) indeterminate. It would be important to know with regard to platform performance.	Thank you for this suggestion and this has been added to Figure 1 (55 – ID NOW; 18 RT-PCR; 1 both).	Figure 1
Reference 21: This title should include "molecular-based tests" based on PMID 33760236.	We are actually citing a somewhat different study (PMID: 35866452). The citation has been updated from the 2021	References

Comment	Response	Changes/Location
	review to 2022, both in terms of the data included in the manuscript and the citation included in the reference section.	

Re: Spectrum01884-24R1 (Accuracy of Point-of-Care SARS-CoV-2 Detection Using Buccal Swabs in Pediatric Emergency Departments)

Dear Dr. Stephen Freedman:

I'm pleased to inform you that your manuscript has been accepted, and I am forwarding it to the ASM production staff for publication. Your paper will first be checked to make sure all elements meet the technical requirements. ASM staff will contact you if anything needs to be revised before copyediting and production can begin. Otherwise, you will be notified when your proofs are ready to be viewed.

Sincerely,
Eleanor Powell
Editor
Microbiology Spectrum